# Detecting Maritime Infrared Targets in Harsh Environment by Improved Visual Attention Model Preselector and Anti-Jitter Spatiotemporal Filter Discriminator

Dongdong Ma, Lili Dong * and Wenhai Xu

School of Information Science and Technology, Dalian Maritime University, Dalian 116026, China
* Correspondence: donglili@dlmu.edu.cn

**Abstract:** Without any prior knowledge, it has always been a serious challenge to accurately detect infrared targets under a maritime harsh environment (MHE). To solve this problem, the main contribution of this paper is to use the improved visual attention model (VAM) preselector and anti-jitter spatiotemporal filter (ASF) discriminator to detect infrared targets in the MHE. The proposed method consists of image preprocessing, the single frame detection of suspected targets and a multi-frame judgment of real targets. First, in the process of single-frame image processing, a combination of the Gaussian difference filtering and local minimum filtering is applied to overcome the uneven background brightness distribution and improve the saliency of the target. Second, an intensity standard deviation method is designed to determine the unevenness of the background. According to the difference in background smoothness, an appropriate "center-surround difference" operation is selected to suppress sea wave interference, and the single frame suspected target candidate region is obtained. Third, in order to "align" the multi-frame image sequence, a method for correcting the position of the inter-frame jitter is proposed. The inter-frame jitter is measured and compensated by the inter-frame block matching results. Finally, according to the three assumptions of multi-frame spatiotemporal filtering, single-frame false targets are filtered out and combined with the OTSU method to segment the real target area. To evaluate the performance index of the proposed method by comparing it with the four other state-of-the-art methods for dealing with the MHE. The experimental results show that the algorithm achieves the maximum detection rate (DR) on the premise of being far lower than the false alarm rate (FAR) of the comparison method. The final experimental results also confirmed that the proposed algorithm is more suitable for infrared target detection in diverse MHEs.

**Keywords:** infrared target detection; visual attention model; multi-frame spatiotemporal filtering; image segmentation

## 1. Introduction

Stable environmental adaptability has always been the bottleneck problem of the infrared search and tracking (IRST) system in the maritime rescue and surveillance process. Therefore, the research on the maritime infrared target detection algorithm under different severe sea conditions is an urgent task. However, the research on infrared target detection methods under the MHE is not mature. The backlight environment is the most common and unavoidable. Once the target on the sea is just between the sunlight and the infrared camera, the backlight infrared image will be captured. Due to the effect of sunlight and waves, compared with ordinary images, there is an uneven background brightness distribution, resulting in high significant highlight noise points. In addition, the saliency of the target will be weakened and the background interference will be enhanced, which makes it more difficult to detect the target accurately, resulting in a lower search and rescue rate. The purpose of this paper is to design a target detection algorithm with a low false alarm rate, a high detection rate, which is not time consuming and has a strong robustness.

With the development of infrared imaging technology, infrared target detection technology has attracted more and more attention from researchers. For example, the infrared target detection methods in the spatial domain mainly include the local contrast method [1,2], which requires the target to be highly significant in the local area, and the maritime backlight image analysis finds that the target local contrast characteristic is not prominent. The multiscale fuzzy metric method [3,4] is to eliminate a large amount of background clutter and noise through a fuzzy metric, and a simple adaptive threshold is selected to segment the target. This method is susceptible to missed alarms caused by high bright spots and a strong background interference. Infrared target detection methods in the frequency domain mainly include the wavelet transform method [5,6], which uses a wavelet basis to convolve the original image to obtain and transform coefficients in different directions, and the application target has strong high-frequency information in multiple directions to filter out the background. The spectral residuals method [7] assumes that the image information is composed of prior knowledge information and significant part information. The logarithmic spectrum of the salient part is obtained by subtracting the information of the prior knowledge from the information of the logarithmic spectrum image, and then the Fourier transform of the image is applied to remove the redundant background, and finally, the saliency map is segmented to obtain the target. The method is suitable for the detection of small and weak targets under a strong background. When the target area is large, it is easy to filter out the real target. The method does not have a strong robustness or stability. In the spatiotemporal domain, infrared target detection methods mainly include the spatiotemporal saliency approach [8,9] and pipeline-filtering algorithm [10,11]; these methods rely on extracting target information from continuous frames. However, none of these methods can resist the interference caused by the jitter between frames. The low-rank tensor completion theory [12,13] considers that the inherent spatial correlation between image pixels indicates that the background is in a continuous manner and highly correlated. On the contrary, the targets are regarded as the object of breaking the local correlation. Therefore, the segmentation of the target from the background can be regarded as the recovery of the low-rank matrix. The low-rank tensor completion theory mainly includes methods based on the low rank matrix recovery theory. At first, Gao proposed the IPI [14] model, then Guo proposed the ReWIPI [15] model and Zhang proposed the NRAM [16] model. Based on the dictionary representation method, Li initially proposed the LRR [17] model, then Lu proposed the OLDSBD [18] model and Zhang proposed the SRWS [19] model. Based on the tensor recovery theory and method, Dai extended the IPI to the IPT [20] model, Zhang proposed the ECASTT [21] model and Kong proposed the NTFRA [22] model. The disadvantage of these methods is its poor robustness to complex environments with a lot of noise. Obviously, this method is difficult to apply in a backlight image. At present, deep learning [23,24] has once become a hot topic. For example, Gao constructed an infrared target detection neural network based on low-dimensional and high-dimensional feature matching [25]. Zhao proposed an infrared small target detection based on the Generic Countermeasures Network (GAN) [26]. Dai implanted the traditional local contrast into the deep network to achieve infrared target detection [27]. However, the method has extremely high requirements on the quality and quantity of the training samples. If the limited training sample is not enough to contain more information, the target will not be accurately distinguished. Additionally, deep learning methods will consume more time. The VAM [28,29] is a computer model based on the human visual perception system. The model can successfully focus on the region of interest when the human eye is observing a particular scene. In the actual process of searching and detecting the marine target, the target area is generally also a prominent area that easily attracts visual attention. Compared with the above algorithm, the main advantage of this model is that it adopts a multi-feature processing method, which can detect the target with a large difference of gray and contrast at the same time; also, the "center-surround difference" operator can highlight different significant information, so that the required information can be extracted selectively. The method integrates multiple

feature channels; we can extract the corresponding feature map by analyzing the backlight image features. The main contributions of this paper are as follows:

(1) The orientation feature, contrast feature, infrared thermal feature and frequency domain feature of the target are analyzed, a preprocessing method combining the Gauss difference filter and local minimum filter is constructed to improve the target saliency in the backlight scenario.

(2) An intensity standard deviation method is designed to distinguish the background non-uniformity; the purpose is to select a suitable "center-surround difference" operator to suppress the interference of highlight points according to the background smoothness. It also reduces the time consumption of a traditional VAM.

(3) According to the space–time characteristics of the maritime target, three hypotheses are proposed to realize the multi-frame space–time filtering of the image, to further eliminate the strong background and sea clutter interference.

(4) Considering that the imaging system will produce an inter-frame jitter in the process of practical application, this paper proposes a method of inter-frame jitter position correction to "align" the image sequence, so as to improve the real target detection rate.

The remaining part of this paper is organized as follows. In Section 2, the characteristics of backlight maritime infrared images are analyzed and described, and a preprocessing method to improve the target saliency is proposed. The suspected target pre-screening method and the real target judgment strategy is introduced in detail. In Section 3, we first introduce the experimental data set, then verify the effectiveness of the improved strategy and carry out a comparative experiment. Finally, the limitations of the proposed method are given. An overall conclusion is drawn for a summary of this paper in Section 4

## 2. Materials and Methods

In this section, we analyze the characteristics of backlight maritime infrared images and propose a preprocessing method that can improve the saliency of the target.

### 2.1. Image Feature Analysis

Image feature analysis is the key step to achieve an accurate detection of the target. The primary purpose is to find the difference between the target and the background in the MHE. This paper mainly analyzes the typical infrared maritime image in Figure 1 from the image's direction features, contrast features, infrared thermal features and frequency domain features.

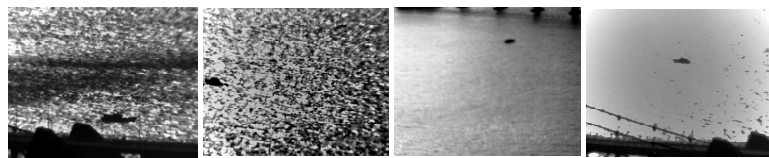

**Figure 1.** Typical infrared maritime backlight image.

First, calculate the local orientation characteristics of the entire image by Equations (1) and (2), as shown in Figure 2. Where the $G_x$ and $G_y$ represent the horizontal and vertical gradient amplitudes calculated by the Sobel operator, respectively. $\beta$ represents the compensation angle. $\theta$ represents the gradient distribution. The results show that the target has no unique orientation feature compared with the background.

$$\beta = \begin{cases} 0°, & \arcsin G_x > 0 \text{ and } \arcsin G_y > 0 \\ 180°, & \arcsin G_x < 0 \\ 360°, & \arcsin G_x > 0 \text{ and } \arcsin G_y < 0 \end{cases} \quad (1)$$

$$\theta = \arctan \frac{G_y}{G_x} + \beta \quad (2)$$

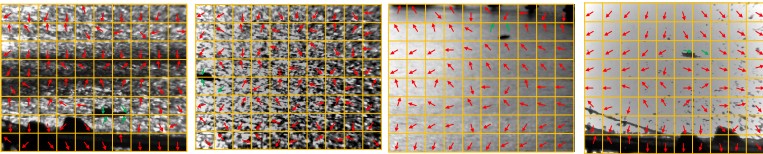

**Figure 2.** Orientational gradient distribution diagram.

Second, the contrast of the entire image is calculated by Equation (3): $g_f g_f$ represents the average gray level of the foreground and $g_b g_b$ represents the average gray level of the background. For the convenience of the observation, the competitive area with the same or a higher local contrast as the target area is marked as green, as shown in Figure 3.

$$\text{Contrast} = \frac{max\left[g_f, g_b\right]}{min\left[g_f, g_b\right]} \quad (3)$$

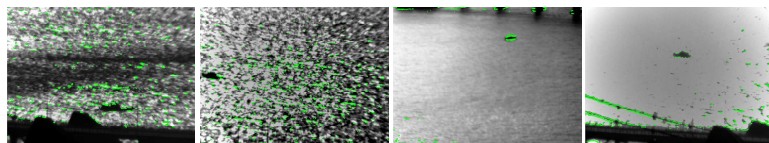

**Figure 3.** Infrared image feature analysis and preprocessing method under MHE.

It can be found that strong backgrounds such as sea waves and bridges also have strong contrast characteristics in a local area, and the target does not have the strongest local significance. Third, from the infrared thermal map calculated in Figure 4, the backlight image has an uneven background brightness distribution due to the influence of the sea waves and sunlight, and the target contour and edge information are seriously damaged. A further observation found that the overall target is at the lowest gray level.

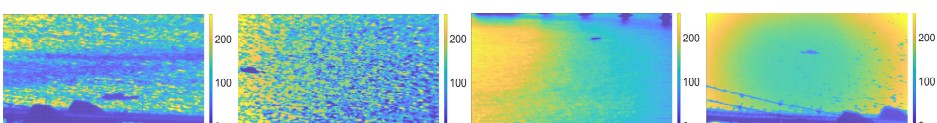

**Figure 4.** Infrared thermal image of maritime target in backlight environment.

Finally, this article analyzes from the perspective of the frequency domain and applies Fourier transform and inverse Fourier transform technology to process the backlight maritime infrared image. The processing result is shown in Figure 5. The (a1–d1) is high-frequency information and (a2–d2) is low-frequency information.

We can find that the strong background is at the lowest frequency position and the noise, such as sea waves, is at the highest frequency position.

### 2.2. A preprocessing Method to Improve Target Intensity

According to the feature analysis, the paper proposes a preprocessing algorithm that can improve the saliency of the target. The overall process is shown in Figure 6.

First of all, according to the gray-level feature analysis results, the target is at the lowest gray level. Secondly, according to the results of the frequency domain feature analysis of the image, it is possible to highlight the saliency of the target by filtering out the highest and lowest frequency information, and a Gaussian difference filter is constructed. Finally, the local minimum image and the Gaussian difference filtering result image are multiplied by Equation (4) to generate the required preprocessed image that can improve the saliency of the target. The *ORI* represents the original image, *Min_Conv* and *GS_Conv* represent the

local minimum kernel and Gaussian convolution kernel, respectively, *PRI* represents the preprocessed result image and the preprocessed result image is shown in Figure 7.

$$PRI = |ORI \otimes Min\_Conv|_{\geq 0} \times |ORI \otimes GS\_Conv|_{\geq 0} \tag{4}$$

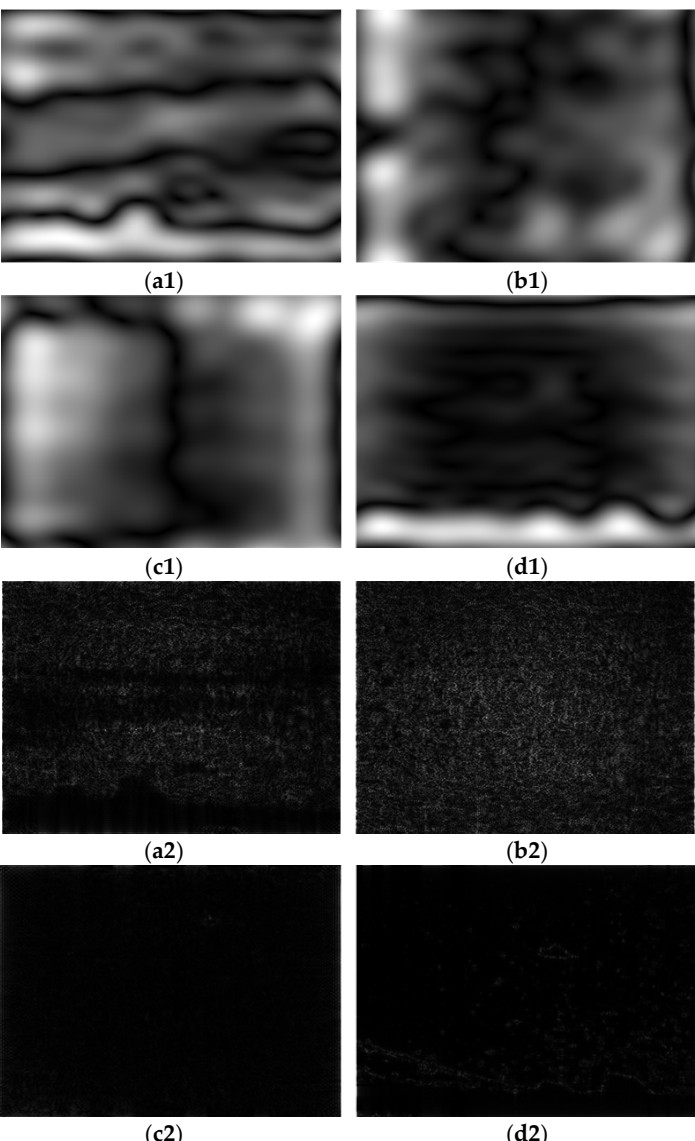

(a1)  (b1)

(c1)  (d1)

(a2)  (b2)

(c2)  (d2)

**Figure 5.** Frequency domain characteristic analysis. (**a1–d1**) High frequency information. (**a2–d2**) Low frequency information.

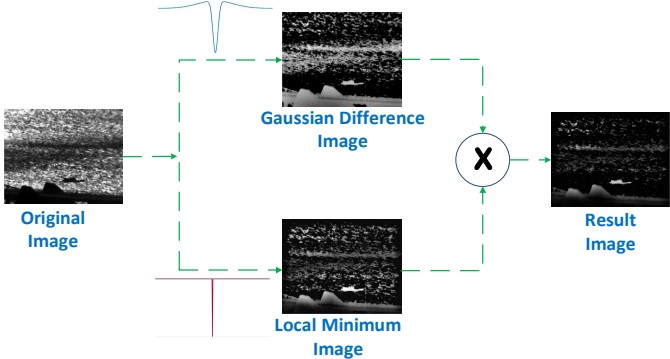

**Figure 6.** Flowchart of preprocessing algorithm to improve target saliency.

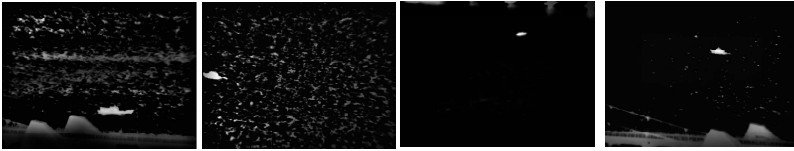

**Figure 7.** Preprocessed results of images in Figure 1a–d, respectively.

It can be found that the target basically occupies the highest gray value from Figure 7, and the intensity and quantity of the sea wave noise and strong background noise are significantly reduced.

In Figure 8, it is found that the number of competitive regions with the same or higher local contrast with the target region is significantly decreased after preprocessing. Therefore, the preprocessing method is more beneficial for subsequent target detection.

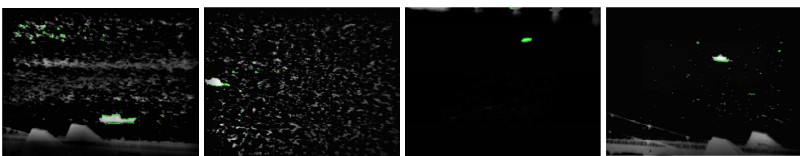

**Figure 8.** Competitive area with the same or higher local contrast as the target area after preprocessing.

At present, there are many computational VAMs in the field of optical image processing, and these models have been successfully applied in target detection [30–32]. Based on the improved VAM preselector and ASF discriminator, this paper proposes an algorithm for infrared target detection under the MHE. The overall structure and process of the proposed algorithm are shown in Figure 9, which is mainly divided into a suspected target pre-screening stage and a real target judgment stage.

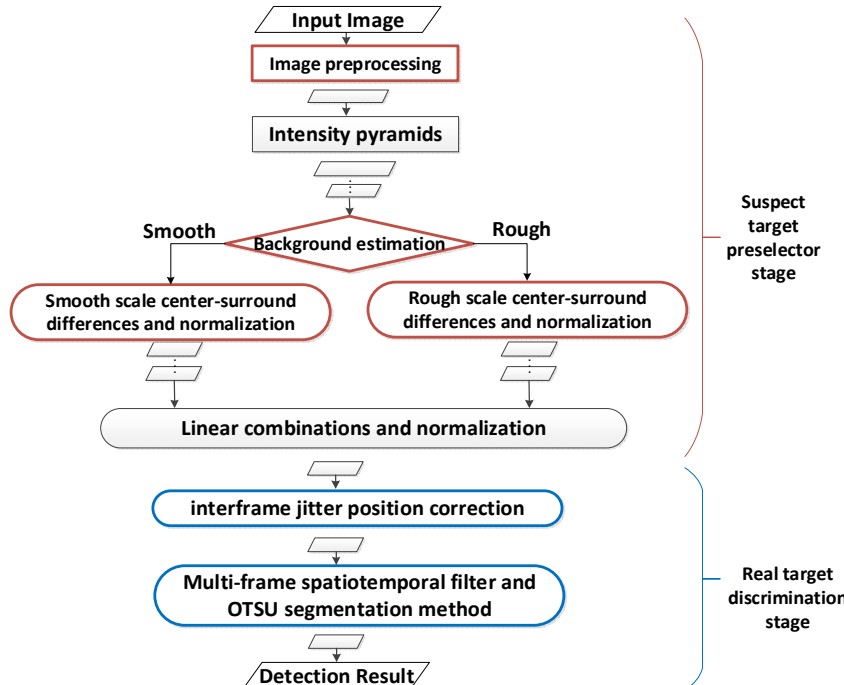

**Figure 9.** Overall structure and flow of the proposed algorithm.

### 2.3. Suspected Target Pre-Screening Stage

First, according to the analysis results in Section 2.1, the target and the background have similar orientation features, so the orientation pyramid in the VAM is abandoned to decrease the clutter interference and reduce the complexity of the algorithm. Based

on the characteristics of the target with a stronger local contrast and global gray after preprocessing, the intensity feature is selected to construct the pyramid. Second, the "center-surround difference" operation has always been a crucial step of the VAM. The core is to perform the difference processing at different scales, and the purpose is to achieve feature extraction. The traditional VAM method is to define the center scale *c* (*c* = {2,3,4}) and the surround scale *s*, where *s* = *c* + *σ* (*σ* = {3,4}). The result of processing the data set in Figure 1 is shown in Figure 10. It can be found that the saliency of the target under different scale operations is not the same. For the data sets (a) and (b), the "center-surround difference" operation results of the 4–7 and 4–8 layers make the target have the strongest significance, as shown in the green box in Figure 10. However, the results of the 2–5 and 2–6 layers in data sets (c) and (d) make the target the most significant, as shown in the red box in Figure 10. Therefore, a different "center-surround difference" operation should be adopted for infrared images in the MHE. Further analysis found that the data sets (a) and (b) are more unevenly distributed in the local area than the data sets (c) and (d). Therefore, the corresponding "center-surround difference" operation can be selected according to different background intensities to reduce the background interference. The gray scale varies dramatically at multiple locations with non-uniformity in Figure 1a,b. On the contrary, the slow and uniform grayscale change in Figure 1c,d. The intensity standard deviation can effectively characterize the degree of heterogeneity of the background. To avoid the impact of the target and the strong background (bridge), our designs for a random intensity standard deviation method to determine the non-uniformity of the background are shown in Equation (5).

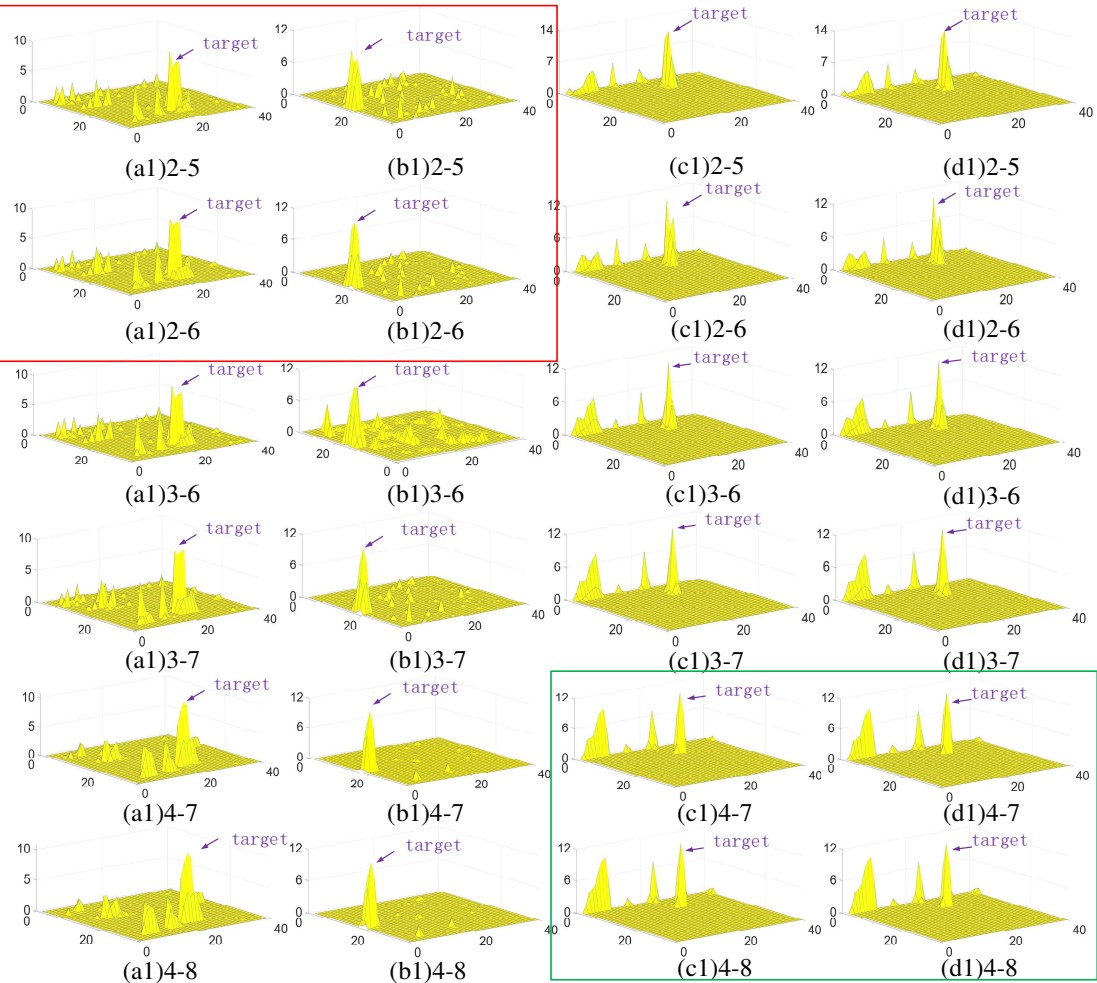

**Figure 10.** Multi-scale "center-surround difference" analysis results.

$$RISD = \sqrt{\frac{(\sum_{i=1}^{k}(S_i) - max(S_i) - min(S_i))^2}{k}} \tag{5}$$

where $k$ represents the number of random samplings of the block regions, $S_i$ is the local standard deviation of each block and *RISD* is the random intensity standard deviation of the entire image. More than 20,000 maritime infrared images under different sea wave intensities are selected, and the statistical results obtained are shown in Figure 11. It can be seen from Figure 11 that the intensity standard deviation of (a) and (b) is significantly higher than that of (c) and (d). The maximum intensity standard deviation of the smooth background is 0.1111.

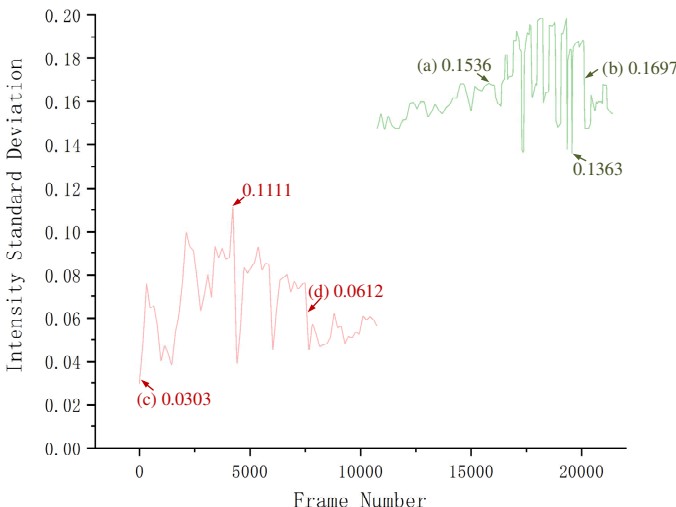

**Figure 11.** Intensity standard deviation under different sea wave background.

In contrast, the minimum standard deviation of the intensity of the non-uniform background is 0.1363. Therefore, it can be found that the intensity standard deviations of the two categories of images are significantly different. The selected threshold is the average of the maximum standard deviation of the smooth background and the minimum standard deviation of the rough background (threshold 0.1237). Experiments have confirmed that it has the best experimental effect.

*2.4. Real Target Judgment Stage*

Figure 12(a1–d1) shows the saliency map obtained by applying the VAM proposed by ITTI; (a2–d2) is the saliency maps obtained after the suspected target screening stage. It can be found that the proposed above method significantly reduces the interference of sea waves noise, and also attenuates the saliency of some backgrounds. However, further observation shows that some background interference is still not eliminated after the above-improved method. To solve this problem, this paper proposes a method based on the inter-frame jitter position correction and multi-frame spatiotemporal filtering to judge the suspected target and get the real target. Our proposed three hypotheses are based on the temporal and spatial characteristics of the target.

(1) The target shift between adjacent frames will not be too large;
(2) The intensity of the target in the saliency map of adjacent frames does not differ greatly;
(3) The target will appear continuously within a certain time frame.

When one of the above three hypotheses is not satisfied, it is judged as the clutter interference. Only when the above three hypotheses are satisfied at the same time, we judge it as the real target, as shown in Figure 13.

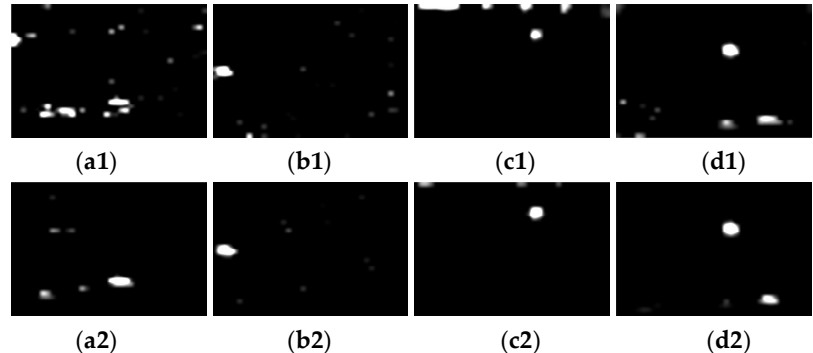

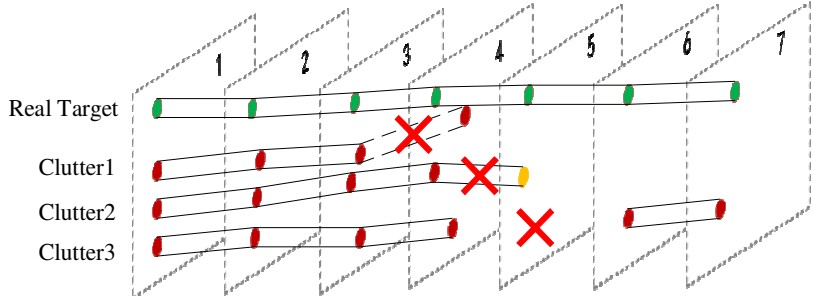

**Figure 12.** (**a1**–**d1**) are the saliency maps obtained by the VAM method proposed by ITTi, (**a2**–**d2**) are the saliency maps after applying the above improved method.

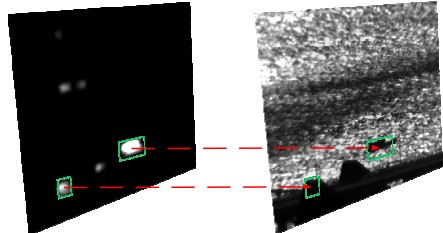

**Figure 13.** Three cases of false targets based on spatiotemporal characteristics.

Considering that in the actual sea surface target search task, the slight jitter of the imaging system will cause a strong frame jitter. However, the commonly used anti-shake measures are difficult to ensure the complete elimination of the jitter between frames [33–35], so it is easy to cause an excessive shift of the target between adjacent frames due to the vibration of the imaging platform, resulting in a missed detection. Therefore, before applying the multi-frame spatiotemporal filtering method, this paper proposes a method of inter-frame jitter position correction, which measures the jitter of adjacent frames through the inter-frame block matching results of the former frame and the current frame. The detailed steps are as follows:

Step one: find the two most significant regions from the saliency map generated in the current frame; the corresponding position of the original image is the block to be matched, as shown in Figure 14.

**Figure 14.** Determine the location of the block to be matched in the original image.

Step two: calculate the highest normalized cross-correlation matching coefficient *NCC* between the two block images of the current frame and the former frame, as shown in Equation (6). $C$ represents the current frame data, $F$ represents the former frame data and $\overline{C}$ and $\overline{F}$ represent the average value of the corresponding data. The best matching position is the pixel position of the maximum *NCC* calculated from the whole image. The difference between the best matching position and the position of the block images of the current frame is inter frameshift at this time. If the inter frameshift of the two blocks to be matched

is highly similar, it is regarded as valid data, and the shift is recorded as $I\vec{J}Cm$ and $I\vec{J}Cs$, respectively. Once the similarity is not high, the inter-frame jitter shift generated by the blocks with a lower similarity is discarded.

$$NCC(x,y) = \frac{\sum_{m,n}\left[C(m,n) - \overline{C}\right]\left[F(x+m, y+n) - \overline{F}\right]}{\sqrt{\sum_{m,n}\left[C(m,n) - \overline{C}\right]^2\left[F(x+m, y+n) - \overline{F}\right]^2}} \tag{6}$$

Step three: once the difference between the horizontal and vertical inter-frame jitter shift of the image to be matched is less than a certain number of pixels, it is judged as valid data with a low similarity, and the average inter-frame jitter shift $A\vec{I}JC$ of the entire image is calculated, as shown in Equation (7).

$$A\vec{I}JC = \begin{cases} \frac{I\vec{J}Cm + I\vec{J}Cs}{2}, I\vec{J}Cm - I\vec{J}Cs \leq (5,5) \\ 0, \ I\vec{J}Cm - I\vec{J}Cs > (5,5) \end{cases} \tag{7}$$

After compensating for the interframes jitter, the sequence image is realigned and the multi-frame spatiotemporal filtering is realized according to the three hypotheses of the target space–time characteristics. This paper applies Equation (8) to develop hypothesis (1), where $\vec{LF}$ is the centroid position vector of former frame saliency image, and $\vec{LC}$ is the centroid position vector of the current frame saliency image; *MaxShift* is maximum matching frameshift.

Hypotheses (2) is developed by Equation (9), *Norm* is the normalized function, *F(m,n)* is the gray value of the centroid position of the saliency map in the former frame and *C(m,n)* is the gray value of the centroid position of the saliency map in the current frame. If the suspected target can satisfy the above two hypotheses and achieve continuous inter-frame matching, it is judged as the real target and the OTSU method is applied to segment the target area. Otherwise, it is judged as a false target and the gray value of the current area is cleared. Figure 15 shows seven saliency images selected from a video sequence after an inter-frame jitter position correction. The target area is marked with a green frame, and the two clutter is marked with red and blue, respectively.

$$\left\|\vec{LF} - A\vec{I}JC - \vec{LC}\right\|^2 \leq 5\sqrt{2} \tag{8}$$

$$Norm[F(m,n) - C(m,n)] \leq 0.1 \tag{9}$$

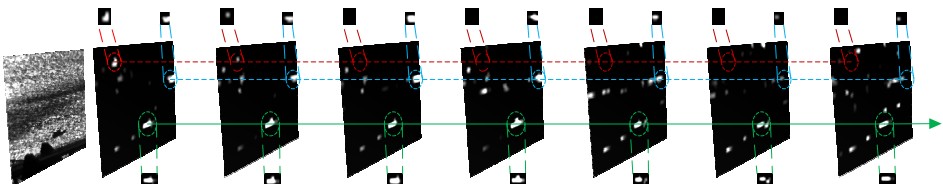

**Figure 15.** Determine the location of the block to be matched in the original image.

First of all, it can be found that the target has always had a strong saliency, and the intensity of the target has been maintained at 0.94–0.96. Second, the target has a small amount of inter-frame shift, which is about one to two pixels. Finally, the target appears continuously in the six saliency images, so it can be determined as the target if the above three assumptions are satisfied. For the clutter marked by a red line, it keeps a similar saliency in the first two frames and disappears suddenly in the third frame, so it cannot achieve the specified matching times, and finally the suspected target is determined as the background interference. For the clutter in the blue line, it maintains a strong saliency in the first five frames, basically at 0.89–0.92. However, at the sixth frame, the significance

suddenly decreases to about 0.58, which exceeds the set experience threshold of 0.1, so it is determined as the clutter interference.

In summary, the novelty in the scene and method is the VAM preselector (consists of image preprocessing and suspected target prescreening) and an ASF discriminator (consists of real target judgment), a strategy which can first effectively improve the significance of the target in a single frame image and suppress the intensity of background clutter, then achieve jitter compensation through adjacent frame images, and finally achieve a further filtering of a strong sea wave clutter through multiple frame sequence images. The proposed method realizes the accurate detection of the distressed target under the backlight scenario.

## 3. Results and Discussion

In this section, first, we introduce the data set of the algorithm verification. Second, in order to better support the rationality and effectiveness of the proposed method, this paper presents three key intermediate results that make the proposed method superior to the state-of-the-art algorithms in infrared ocean detection: (1) The preprocessed result image and signal-to-clutter ratio gain *(SCRG)*. (2) The saliency map under a different "center-surround difference" operation. (3) The amount of clutter and an anti-jitter performance. Finally, the advantages of the proposed strategy and the comparison algorithm in the detection results and receiver operating characteristic curve (ROC) are compared.

### 3.1. Real Target Judgment Stage

This paper analyzes and verifies 20,000 maritime infrared images and selects 8 typical images in different scenarios for display, as shown in Figure 16. Table 1 shows the description of the experimental environment and typical image scenes. It can be seen that the data set is collected by infrared cameras from different bands under different wind speeds and wave strengths. The images include targets of different sizes disturbed by waves, sea antennas, islands and a lens flare. Therefore, the data set includes the common challenges of target detection in the MHE, and the good detection performance of these data sets can verify the effectiveness of the algorithm.

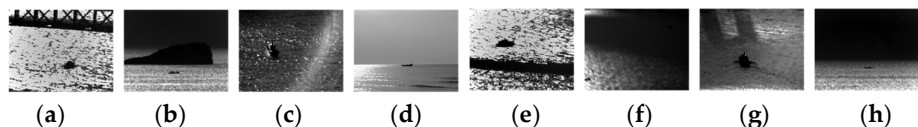

(a)　　　(b)　　　(c)　　　(d)　　　(e)　　　(f)　　　(g)　　　(h)

**Figure 16.** (**a**–**h**) is a representative images of each image sequence.

### 3.2. Verify Intermediate Results

In this section, we prove the effectiveness of the proposed three key units, so we show the intermediate results and performance analysis of each unit.

Unit one: the preprocessing algorithm to improve the target saliency.

The purpose of the preprocessing algorithm is to suppress the interference of the background and sea waves, thereby enhancing the saliency of the target. Through the representative image processing results in Figure 17, it can be observed that the target and background gray levels are reversed, and the target has a strong gray level. It can be seen from Figure 17a,b,e that the strong background is suppressed and the ocean wave noise is significantly reduced. The lens flare and background reflections are clearly suppressed in Figure 17c,g. The connected dark or bright lines are also eliminated in Figure 17d,f,h.

In order to better confirm the performance advantages of the proposed preprocessing algorithm, we choose *SCRG* as the evaluation index. *SCRG* is defined as:

$$SCRG = \frac{(S/C)_{out}}{(S/C)_{in}} \tag{10}$$

where *S* and *C* are the average target intensity and clutter standard deviation, respectively. $(.)_{in}$ and $(.)_{out}$ are the original image and the result image. The *SCRG* index measures the magnification of the target relative to the backgrounds before and after processing. The experimental results of the proposed method with the index are shown in Table 2. It can be found that the processed image has a high *SCRG*, which reaches the maximum of 52.7 in the data (f).

**Table 1.** Experimental environment and image description of each image sequence.

| Image Sequence | Number of Test Sets | Experimental Environment and Image Description | | | |
| --- | --- | --- | --- | --- | --- |
| | | Wind Speed (m/s) | Wave Height (m) | Infrared Camera | Background Interference |
| (a) | 1025 | 8.6–9.8 | 1.8–2.0 | Long wave un-refrigeration | Bridge Bright and dark spots |
| (b) | 1342 | 5.0–5.4 | 0.3–0.5 | Medium wave refrigeration | Sea-sky line IslandsSea waves |
| (c) | 2085 | 6.0–7.4 | 0.7–0.9 | Long wave un-refrigeration | Lens flare Highlight ocean noise |
| (d) | 4463 | 2.7–3.0 | 0.1–0.3 | Medium wave refrigeration | smooth streaks |
| (e) | 507 | 8.6–9.8 | 1.8–2.0 | Long wave un-refrigeration | Connected dark lines |
| (f) | 768 | 5.0–5.4 | 0.3–0.5 | Medium wave refrigeration | Shadow bright and dark spots |
| (g) | 821 | 2.0–3.4 | 0.1–0.3 | Long wave un-refrigeration | Bridge reflection |
| (h) | 2017 | 5.0–5.4 | 0.3–0.5 | Medium wave refrigeration | Sea-sky line Sea waves |

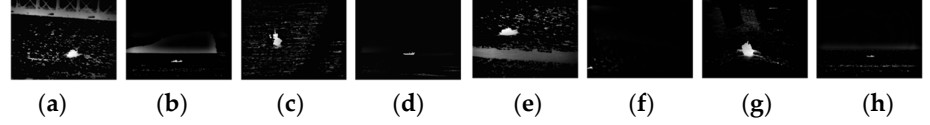

　　(a)　　　(b)　　　(c)　　　(d)　　　(e)　　　(f)　　　(g)　　　(h)

**Figure 17.** (**a–h**) is preprocessed results of images in Figure 16a–h, respectively.

**Table 2.** *SCRG* value after image preprocessing.

| Data | (a) | (b) | (c) | (d) | (e) | (f) | (g) | (h) |
| --- | --- | --- | --- | --- | --- | --- | --- | --- |
| *SCRG* | 24.6 | 25.4 | 23.5 | 36.0 | 26.7 | 52.7 | 21.6 | 50.1 |

Unit two: the different scales "center-surround difference" operation.

For the background with different smoothness, the "center-surround difference" operation under different scales can highlight the significance of the target. Figure 18(a1)–(h1) is the feature map obtained by the "center-surround difference" operation applied in the ITTI VAM, and (a2)–(h2) is the result of the selecting corresponding scales "center-surround difference" operation. It can be found that (a2), (b2), (e2), (h2) has less clutter interference than (a1), (b1), (e1), (h1), and the clutter intensity is also obviously suppressed. Compared with (c1) and (g1), the target area of (c2) and (g2) is well preserved, which contributes to the integrity of the target contour in the final detection result.

Unit three: interframe jitter position correction.

Inter-frame jitter compensation is a key step in realizing multi-frame spatial filtering. Only by matching multi-frame image sequences can the target appear the strongest continuous saliency. To verify the performance of the proposed inter-frame jitter position

correction method, the mean square error (*MSE*) of the overlapped saliency map after adjacent frames position correction is calculated, as shown in Equation (11).

$$MSE = \sqrt{\frac{\sum_{x=1}^{W}\sum_{y=1}^{H}(Sf(x,y) - Sc(x,y))^2}{W * H}} \tag{11}$$

*W* and *H* represent the width and height of the image, respectively. *Sf* and *Sc* represent the saliency map of the former frame and the current frame, respectively. Since the adjacent frames are acquired in a short period time (our infrared imaging camera is set to 40 frames per second), their saliency maps should be similar to each other. Once dislocation is caused by the jitter between the frames, the *MSE* between them should be amplified. Figure 19 shows all the correction results of the eight image sequences, including the adjacent frame *MSE* value before the correction, the amount of inter-frame correction in the horizontal and vertical directions and the *MSE* values of the adjacent frames after applying the proposed correction method. It can be found that the *MSE* value after correction is smaller than before correction, especially in the position where the jitter is relatively large, as the maximum *MSE* value before correction reaches 0.9 and after the horizontal and vertical position correction, the *MSE* value drops to about 0.2. This proves the effectiveness of the proposed image correction method and achieves the purpose of aligning the image sequence.

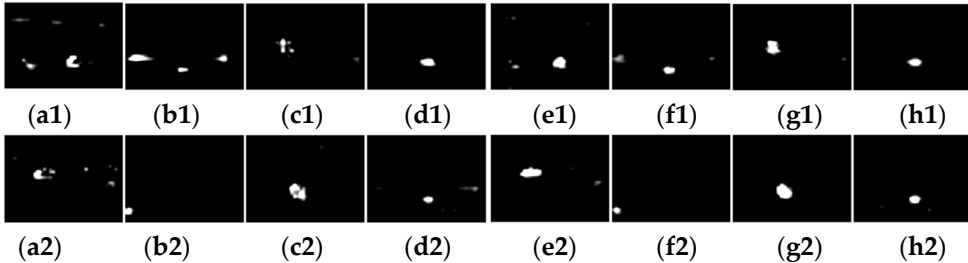

**Figure 18.** (**a1**–**h1**): operation results of "center-surround difference" applied in ITTI model. (**a2**–**h2**): the results of the "center-surround difference" operation of different scales in our model.

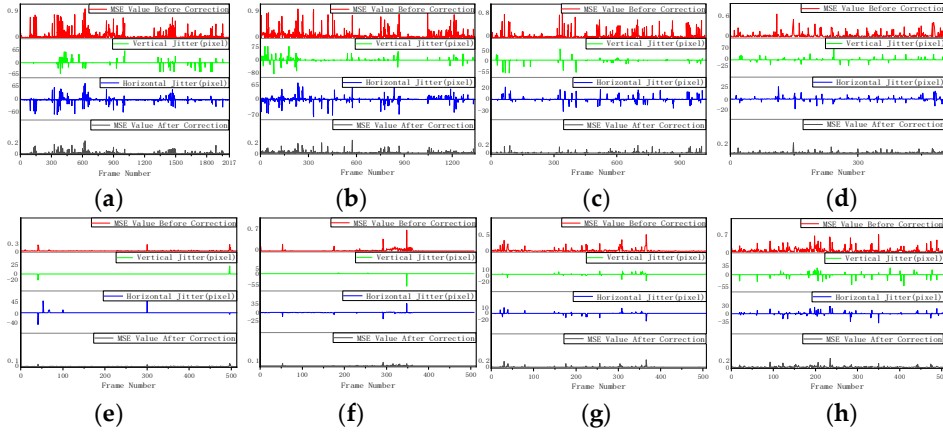

**Figure 19.** (**a**–**h**) is verification results for inter-frame jitter position correction.

## 3.3. Comparison of Experimental Results

This paper selects four marine target detection algorithms to compare with the proposed algorithm. These algorithms are aimed at the target detection, which are the absolute directional mean difference (ADMD) method [36], partial sum of the tensor nuclear norm (PSTNN) method [37], the novel local contrast descriptor (NLCD) [38] and the multiscale tri-layer LCM (TLLCM) method [39]. The final detection results of each algorithm are compared in Figure 20 (we performed morphological dilation on the results to facilitate observation). The proposed algorithm can overcome strong background interferences such

as islands, bridges and connected dark lines, as shown in Figure 20a,b,e. At the same time, it can also be adapted to the sea–sky environment, as shown in Figure 20b,d,h. It can accurately suppress the interference of sea waves with high-brightness characteristics, effectively reducing the generation of false targets, as shown in Figure 20a–c,e,f,h. In addition, when the lens flare and bridge reflection appear in the image, as shown in Figure 20c,g, the proposed algorithm can still accurately locate the target area. These data analysis results also validated the effectiveness of the proposed strategy to detect large and small targets. The ADMD method can accurately detect targets in multiple scenes, the reason is that the ADMD method is based on directional information; false alarms may appear in the strong background area close to the target characteristics, as shown in Figure 20a,b,e,h. The PSTNN method can well eliminate the interference of a strong background and perfectly compensate for the disadvantages of the ADMD algorithm. However, the method to stably detect the target mainly depends on the whether the image satisfies the target sparsity and background low-rank assumptions. The local complex scene will lead to more false alarms, as shown in Figure 20c,f,g,h. For the NLCD method, the backlit image itself has a stronger interference than the other scenes. This method is more sensitive to pixel-sized noises with a high brightness (PNHB), so it produces a large number of missing alarms. The NLCD method is only suitable for backlight scenes with weak ocean waves and strong target textures. For the TLLCM method, the core of the method is to estimate the background so as to filter out the background and extract the target. It can be found that the detection effect is not ideal from the results, only when the target is accurately detected in Figure 20c,g. However, there will be varying degrees of false alarms and missed alarms for other data sets. The reason is that the size of the target is different. Once the background is not estimated, the target will be filtered out.

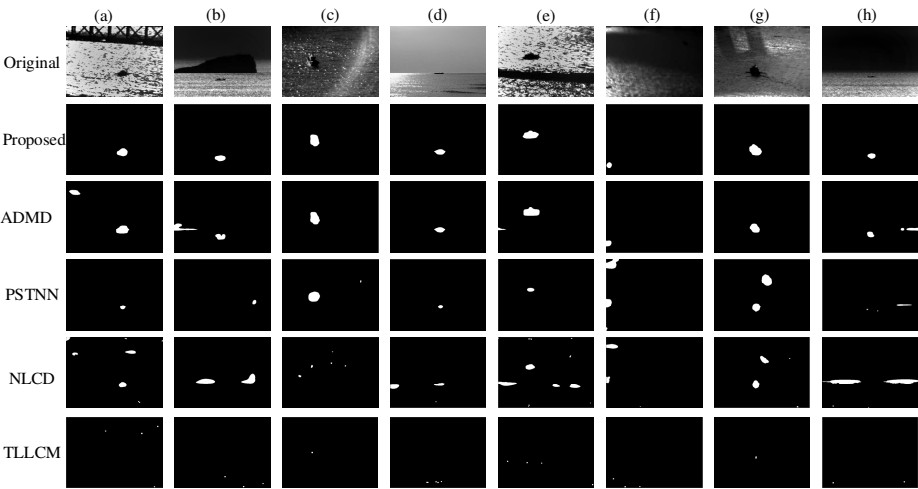

**Figure 20.** (**a**–**h**) Comparison of detection results of six detection algorithms.

The Gaussian difference, local minimum detection and multi-scale construction in the VAM applied in this paper are all based on a convolution calculation. When $N$ represents the total number of image pixels and $p$ represents the local neighborhood scale, the time complexity of the proposed algorithm is $O(p^2 N)$. The time complexity of other methods has been calculated in their respective literatures. The time complexity and time consumption of all methods are finally shown in Table 3, from which we can find that the time complexity of this algorithm is the lowest. We simplified the traditional VAM, mainly reflected in: (1) based on the analysis results of the image features in Section 2.1, we abandoned the construction of the orientation pyramids. (2) Based on Section 2.3, this paper uses some, but not all, intensity pyramids to perform the "center-surround difference" operator. Therefore, the above two schemes also reduce the time consumption of the traditional VAM, making the proposed methods meet the requirements of real-time processing. In contrast, other

methods have the characteristics of a high complexity and long time consumption, so they are difficult to meet the engineering application.

**Table 3.** The complexity and time consumption of typical methods.

| Method | Proposed | ADMD | PSTNN |
|---|---|---|---|
| Complexity | $O(p^2N)$ | $O(p^2N^3K)$ | $O(n_1n_2n_3\log(n_1n_2) + n_1n_2{}^2[(n_3 + 1)/2])$ |
| Time/s | 0.082 | 0.185 | 0.158 |
| Method | NLCD | TLLCM | |
| Complexity | $O(\xi p^6N)$ | $O(LR^2\log R^2MN)$ | |
| Time/s | 0.281 | 4.661 | |

In order to further prove the performance advantages of the proposed algorithm, we compared the ROC curves of DR and FAR of each algorithm, as shown in Figure 21.

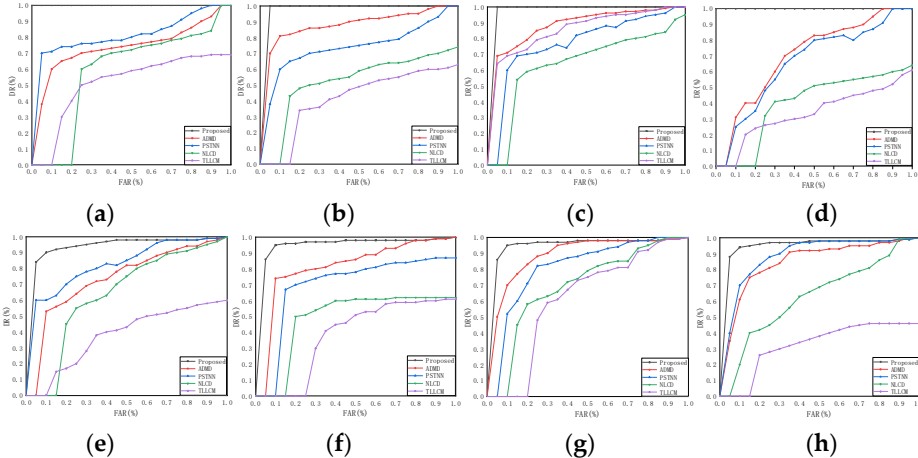

**Figure 21.** (**a**–**h**) is ROC curves of detection results of eight real sequences.

It can be observed from Figure 21 that the ROC curve of the proposed method reaches its maximum DR when it is much lower than the FAR of the comparison algorithm. In contrast, the comparison algorithm can only achieve the maximum DR under a higher FAR, and not every type of data set can achieve a 100% DR. The TLLCM method has the worst performing index. The reason is that it is designed according to the local characteristics of the target and the background. However, targets submerged by strong sunlight usually lose their local characteristics. Therefore, it performs poorly in this situation. Although the PSTNN method can accurately detect some targets, it does not use the temporal and spatial information of the targets to further reduce false alarms. In this way, the maximum DR can be reached in the case of a large FAR. When the ocean waves are small and the target has strong texture features, NLCD can have a high detection accuracy. However, once the sea waves are large and accompanied by a strong background interference, the DR will be severely reduced. The ROC curve of the ADMD method can achieve a higher DR than other comparison algorithms, but the problem exposed is that the strong background area close to the target characteristics cannot be filtered out during the detection of the image sequence. If we define the best performance as achieving the maximum DR while reducing the FAR as much as possible, this will verify that the proposed method is most suitable for detecting infrared targets in the MHE.

### 3.4. Limitations

Due to the fact that the contrast, gradient and intensity characteristics of the target in the backlight environment are different from the ordinary environment, the proposed method is more suitable for target detection in the backlight environment. At present, we

are solving the problem of accurate image classification in the backlight environment and general environment, to better serve engineering applications.

## 4. Conclusions

This paper's major contribution is to provide an effective improved VAM preselector and ASF discriminator strategy for effectively extracting targets' saliency from an infrared maritime backlight image and help improve the target detection accuracy. Through an adaptive selection of the appropriate "center-surround difference" operator, the target saliency is enhanced while reducing the saliency of sea waves and the strong background. The experimental results also show that this optimized operation can effectively cut down the number of clutter interference. The inter-frame jitter position correction method used in the process of multi-frame image processing can effectively overcome the vibration of the imaging platform. The experimental results also confirm that the *MSE* value after correction is reduced from a maximum of 0.9 to 0.2. After applying the multi-frame spatiotemporal filtering and OTSU method, the real target is accurately segmented from a single frame of highly suspected targets. The experimental results based on real data sets show that compared with other algorithms, the proposed algorithm has a higher DR for infrared marine targets in different wind and wave environments, targets of different sizes and a strong background interference.

**Author Contributions:** Conceptualization, D.M. and L.D.; software, W.X.; All authors have read and agreed to the published version of the manuscript.

**Funding:** This paper was supported in part by the Fundamental Research Funds for the Central Universities of China under Grant 3132019340 and 3132019200. This paper was supported in part by high tech ship research project from ministry of industry and information technology of the people's republic of China under Grant MC-201902-C01. (Funder: Dong, L.).

**Conflicts of Interest:** The authors declare no conflict of interest.

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
