# Peer review of "Detecting Maritime Infrared Targets in Harsh Environment by Improved Visual Attention Model Preselector and Anti-Jitter Spatiotemporal Filter Discriminator"

_remotesensing, doi:10.3390/rs14205213_

Round 1

Reviewer 1 Report

This paper uses the improved visual antention model preselector and anti-jitter spatiotemporal filter discriminator to detect infrared targets in MHE. Some interesting results are provided and the experimental results show the performance of the proposed algorithm. Below are some comments & questions:

1, My main concern is the novelty of this paper. Actually, preprocessing, suspected target pre-screening and real target judgment stages have been discussed in previous papers. The paper claims to provide an effective VAM preselector and ASF discriminator strategy, but I am just wondering what is new here. The authors need to clarify the novelty either in scenario or methods.

2, I would also suggest to change the organization of this paper. As can be seen, the paper is just composed of 4 sections. Regardless to the introduction and conclusion, we just have 2 sections left. The second sections is "related work", so I really don't see what is proposed in this paper. 

3, I would like to suggest to analyze the complexity of the proposed method and compare it with the previous methods.

Author Response

Detailed response files are attached.

Reviewer 2 Report

Good for the publication in the present form

Author Response

Thank you for your review.

Reviewer 3 Report

An interesting paper, I have no specific comment : the methodology is clearly presented, and the results on experimental scenes are quite convincing.

Author Response

Thank you for your review.

Reviewer 4 Report

The methods used to find objects in images recorded in infrared are also used in ordinary images. There are no explained differences in the procedure. The algorithms used are also known from applications. I did not notice references to the world literature on the subject in the introduction. I suggest expanding the bibliography. 

Equations 1 and 2 use designations that the authors do not explain, such as "Gx."

Figure 5 is not very clear. The a2-d2 images require additional explanation. 

Section 2.2 is not at the publication level. They are textbook news.

Figure 10 not very legible. 

In my opinion, the work presented is of low scientific quality. There are technical considerations presented in it using applied techniques. I do not see the scientific or research aspect.

Author Response

Detailed response files are attached.

Round 2

Reviewer 1 Report

I am satisfied with the revision.

Reviewer 4 Report

I have no comments